# Dispersion of Aerosols Generated during Dental Therapy

**DOI:** 10.3390/ijerph182111279

**Published:** 2021-10-27

**Authors:** Yusuke Takanabe, Yutaka Maruoka, Junko Kondo, Shotaro Yagi, Daichi Chikazu, Ryuta Okamoto, Masao Saitoh

**Affiliations:** 1Department of Oral and Maxillofacial Surgery, Center Hospital of the National Center for Global Health and Medicine, 1-21-1 Toyama, Shinjuku-ku, Tokyo 162-8655, Japan; ytakanabe@hosp.ncgm.go.jp (Y.T.); jkondou@hosp.ncgm.go.jp (J.K.); shyagi@hosp.ncgm.go.jp (S.Y.); 2Department of Oral and Maxillofacial Surgery, Faculty of Medicine, Tokyo Medical University, 6-7-1 Nishishinjuku, Shinjuku-ku, Tokyo 160-0023, Japan; chikazu@tokyo-med.ac.jp; 3Department of Oral and Maxillofacial Surgery, Graduate School, Tokyo Medical and Dental University, 1-5-45 Yushima, Bunkyo-ku, Tokyo 113-8549, Japan; 4Shin Nippon Air Technologies Co., Ltd., Solutions Division, Hamacho Central Building 8F, 2-31-11 Nihonbashihamacho, Chuo-ku, Tokyo 103-0007, Japan; okamotor@snk.co.jp; 5Center for Medical Education and Sciences, Graduate School of Medicine, University of Yamanashi, Chuo, Yamanashi 409-3898, Japan; masaos@yamanashi.ac.jp

**Keywords:** COVID-19, aerosol particles, dental care, aerosol dynamics, micro engine, super clean laboratory

## Abstract

The novel coronavirus pandemic has resulted in an urgent need to study the risk of infection from aerosols generated during dental care and to conduct a review of infection controls. However, existing studies on aerosol particles related to dental treatment have mainly evaluated only the scattering range. Few studies have been conducted on the specifics of the generation of aerosol particles in clinical settings, their mechanisms and patterns of distribution throughout open or enclosed spaces, the duration that they remain suspended in air, and the amount and size of particles present. To minimize the influence of background particles, laser lights, a high-sensitivity camera, and particle counters were used in a large super clean laboratory to investigate the dynamics of aerosols generated during the operation of dental micromotors. The results indicate that aerosols tend to scatter upward immediately after generation and then gradually disperse into the surroundings. Most of the particles are less than 5 µm in size (only a few are larger), and all particles are widely distributed over the long term. Our research clearly elucidates that aerosols produced in dental care are distributed over a wide area and remain suspended for a considerable time in dental clinics before settling.

## 1. Introduction

The world is currently suffering from a pandemic of severe acute respiratory syndrome coronavirus 2 (SARS-CoV-2, COVID-19) infections. Japan, along with many other countries, is facing an ongoing increase in the number of infected individuals. As a result, there is an urgent need to control the spread of the virus. Many questions remain regarding the infection pathways of COVID-19. It is believed that COVID-19 is mainly spread by droplet-borne transmission or through contact infection; however, airborne transmission is also possible [1,2,3,4,5]. Because the daily tasks of dental practitioners involve examination and treatment in and around the mouth, dental practitioners are in danger, as they can be exposed to saliva or aerosols that contain SARS-CoV-2 [6,7,8,9,10,11]. Insufficient data have been collected regarding the risk of COVID-19 infection among healthcare professionals involved in dental therapy. It has been noted that the use of high-speed rotary cutting tools, ultrasonic scalers, and other frequently used tools in dental care settings can increase this risk [9,11,12,13], as aerosols can permeate wide areas [14].

Past efforts to assess aerosols that are related to dental therapy have only focused on the range of scattering. There are no reports in the literature on how aerosols are generated, how far they spread, how long they drift through the air, how many particles are present, or how large the particles are. Thus, it appears to be of utmost importance, in terms of infection control, to learn more about the dynamics of aerosols that are generated during dental therapy.

It has, however, been reported that many fine particles are typically suspended in the air in dental clinic settings, and it is necessary to accurately investigate the fine particles produced purely by the use of dental treatment equipment [11,15,16]. In this study, laser light and a high-sensitivity camera were utilized to visualize the aerosols that were generated when a high-speed rotating cutting tool (driven by a micromotor) was used in a large super clean laboratory (SCL). Corresponding qualitative and quantitative assessments were performed to measure the quantity of microparticles using particle counters (P/Cs). This report presents new findings on the range, quantity, and persistence of aerosol scattering.

## 2. Materials and Methods

The experiment was conducted in an SCL (specifically, an ISO14644-1 Class 1 cleanroom) for visualization (Shin Nippon Air Technologies Co., Ltd., Chino City, Nagano Prefecture, Japan). A fan filter unit (FFU) was present in the SCL, which made it possible to reduce the number of particles drifting through the air to nearly zero. All the particles measured were aerosols specifically created for this study. The dimensions of the SCL used for the experiment were 4 × 4 m. At the center of this area, a workbench was placed to simulate dental therapy. The workbench had a height of 74 cm, which was assumed to correspond to the natural posture of a worker during dental therapy.

### Aerosol Particle Scattering Experiment

Table 1 lists the materials that were used for the measurements. In the present experiment, all the measurement instruments were first set up and the workers, clad in SCL garments, stood in the middle of the SCL (Figure 1). The FFU was run for 2 min to expel the particles from the SCL until the value on the P/C decreased to 0. Then, the FFU was stopped, and a pause was observed until the airflow stopped.

The devices that were used to generate the aerosols consisted of a portable VIVAace (Nakanishi Inc., Kanuma City, Tochigi Prefecture, Japan) unit and a 1:5 micromotor handpiece X95L (Nakanishi Inc.: four sprays with a maximum rotational speed of 200,000 min^–1^). In a hemispherical bowl that measured 5 cm deep and a circle with an 8 cm radius that simulated the oral cavity, the micromotor was run continuously for 1 min from the height of the rim of the container and in the middle of the container to generate the aerosols. Vacuum suction was not used. The micromotor sprayed water at a rate of 50 mL/min during the task with a rotational speed of 200,000 min^–1^. A transceiver was used to provide cues to the workers from outside the SCL to minimize the generation of dust by worker’s behaviors. The number of aerosols to be scattered to reach equilibrium within 8 min after the task had been confirmed in a preliminary experiment; thus, visualization imaging and particle counting were performed for 9 min.

The steps of our experimental method were as follows:Measurement of the aerosol particles immediately over the table. A visualization coefficient device called Type-S (sensing area: 4 × 20 cm; Shin Nippon Air Technologies) was set up at a location that was 60 cm above the table (position of the head of a worker). The number of particles that were larger than 0.5 µm and passed through was measured continuously for 9 min, and sampling was performed in 1/30 s intervals.Qualitative assessment of the distribution of droplets using a visualization video. Two scanning laser sheet light sources (Parallel Eye H, Shin Nippon Air Technologies) were used to create a 2 × 2 m laser sheet at a location that was 30 cm from the floor. Traversing droplets were monitored continuously for 9 min with a microparticle-visualizing high-sensitivity camera called i-Scope for the distribution of the transiting particles that were larger than 5 µm, and 30 s still images were continuously created. The basic image processing software package ParticleEye Viewer (Shin Nippon Air Technologies) was used for image processing to obtain the visualized accumulated images of the horizontal cross-section. Because the instrument setup and sensitivity issues made it impossible to measure the entire region inside the SCL simultaneously, the SCL was divided into four areas, and the experiment was performed three times each at position. The median values of these three experiments were used in the following data analysis (Figure 1).Quantitative assessment of the droplet distribution by counting. P/Cs (TSI Model 9303 with a suction flow rate of 2.83 L/min, and Particle Plus Model P311 with a suction flow rate of 2.83 L/min) were used to capture the particles. The P/Cs measured the aerosol particles’ size, quantity, and the amount of settled particles over time. Six P/Cs were used in each area and were set up in 50 cm intervals in the 90° direction (1–3) and 45° direction (3–6) at the same height as the workbench (74 cm from the floor). This approach was used to capture the settled particles and to measure the distribution of dispersion. Particles with a grain size of >0.5 μm or >5.0 μm were measured continuously for 9 min, with a sampling duration of 10 s intervals.

## 3. Results

Images captured in preliminary experiments showed that when the micromotor was operated, an ascending airflow was produced, which sent the aerosol particles upward. In the span of a few seconds, the aerosol particles went above the heads of the workers and were close to the ceiling. Most of the large droplets immediately fell instead of rising up. However, the small droplets ascended to the ceiling, spread across the room, and some drifted upward or floated in the air for a considerable time. A similar visualization experiment was also performed on an ultrasonic scaler attached to the same device. It was observed that the amount of scattered fine particles was also small, probably because the amount of water and air ejected was smaller than that of the micromotor. It was found that large particles did not rise to an elevated position prior to the particles settling out of the air. Particles that seemed to be smaller accumulated in a low position, appeared saturated, and then were gradually ejected upward (data not shown).

The Type-S measured the number of aerosol particles released by the micromotor, which was assessed using the integrated value every 0.2 s. There was a peak of 4000–5000 counts/0.2 s or higher, which was measured immediately after the task began. However, the quantity of particles that was measured soon after the commencement of work decreased dramatically to 1/3–1/4, and after 60 s, the quantity of particles stabilized to an average of approximately 200 counts/0.2 s. The number of particles >0.5 µm in size measured in the last 1 min of the 8 min measurement period after the stable state was reached was 57,853 counts/min on average. Thus, a large quantity of aerosol particles was generated (Figure 2).

Parallel Eye H was used to visualize the accumulated images of particles >5 µm in size in a horizontal scattering range. After 30 s, no bright spots were observed on the horizontal cross-section, and the NW direction showed that the particles had been visualized from 60 s onward. They then spread in the SE direction, and they showed a tendency to be somewhat more abundant in the NW and SE directions by 300 s. Subsequently, they were evenly distributed and dispersed in the SCL (Figure 3). 

These particles were captured and observed by the P/Cs. Although the particles >5 µm in size reached a peak at 30–60 s from the start, these particles were low in number and were mostly limited to within 50 cm of the operating sites of the devices. By the end of the experiment, no particles were captured by the P/Cs. In the present experiment, the data were obtained by measuring the number of particles that were less than 5 µm in size. Several particles were measured immediately after the micromotor was started, a peak was reached within 100 cm from the workers, and the particles were centered westward (left of the workers) at 30–90 s from the start. At first, more particles were captured in the NE, but over time, they also gradually spread to the SE, reaching a peak after 210 s. The particles that were captured at each of the points reached equilibrium after approximately 300 s, when they were distributed throughout the room (Figure 4).

## 4. Discussion

Past studies of contamination from aerosols in dentistry have included the use of agar medium to culture oral bacteria. Previous researchers also performed ATP swab testing to study the degree of contamination, a dyeing solution to determine the scattering range, and luminol reaction to investigate blood [12,13,17]. One group also used laser light to assess the spreading of dust generated during cutting of teeth [18]; however, the range was limited to the front of the worker, and the assessment was performed for only 8 s. Several other scholars have reported considerable contamination on the faces and chests of workers as well as on the chests of patients. In addition, few researchers have reported contamination being observed 3 m from the operating sites of devices. All of these studies were focused on the range of scattering of the aerosols. However, how the aerosols are generated in dental therapy, how they are spread, and what size or distribution the particles exhibit have not been previously investigated; thus, the behaviors of aerosol particles have not been clearly explained.

SARS-CoV-2 virus particles are believed to be approximately 0.005–0.2 µm in size [19]. It has been noted that they include aerosol particles that are smaller than 5 µm [20,21], which leaves open the possibility that they are included in the aerosols generated by the exhaled breath of patients during dental therapy [8]. SARS-CoV-2 remained viable in aerosols for 3 h throughout the duration of the experiment, with a reduction in infectious titer from 10^3.5^ to 10^2.7^ TCID_50_ per liter of air. Viruses have a half-life of 1.09 h in aerosols [22]. In this study, particles generated by dental equipment reached a peak after 210 s. The fine particles did not reach zero for at least 9 min. In addition, in the scattering experiment we conducted prior to this experiment, particles were observed to continue settling even after 2 h (data not shown). There is a risk of infection from the air if the saliva of an asymptomatic pathogen carrier is dispersed into the air due to high-speed rotary cutting tools or ultrasonic scalers. Measurements that have been taken in isolation wards or ventilated hospital rooms have shown very low concentrations of SARS-CoV-2 RNA in aerosols. However, they are reportedly elevated in restrooms or frequently congested zones that are used by patients, in which aerosol particles are distributed with various sizes [21], and unmanaged spaces are associated with high risks of COVID-19 infection.

Harrel, et al. described five dental devices and processes known to cause air pollution. Ultrasonic scalers are considered the greatest source of aerosol contamination, and the use of a high-volume evacuator reduced airborne contamination by more than 95%. Bacterial counts indicated that airborne contamination in air polishing was nearly equal to that of ultrasonic scalers, and available suction devices reduced airborne contamination by more than 95%. Bacterial counts also indicated that airborne contamination by air–water syringe was nearly equal to that of ultrasonic scalers, and a high-volume evacuator reduced airborne bacteria by nearly 99%. Tooth preparation with an air turbine handpiece exhibited minimal airborne contamination if a rubber dam was used. The bacterial contamination properties of tooth preparation with air abrasion remains unknown; however, extensive contamination with abrasive particles has been found [23].

In our previous experiments, fine particles generated by the air turbine handpiece were visualized most, followed by the micro engine and then the ultrasonic scaler (data not shown). However, in this study, only the data of the micromotor were obtained. The cause of this discrepancy is unknown, but additional experiments with ultrasonic scalers might answer this question. This study was conducted in a closed space to completely eliminate background particles. Therefore, we did not use any vacuum devices, but our result clarified a detailed aspect of the movement of fine particles in an unventilated space.

Unfortunately, it is impossible to prevent the generation of aerosols during dental therapy completely. It has been reported that using an extra-oral vacuum suctions at most 75% of aerosol particles [24]. By combining an intraoral vacuum and an extraoral vacuum, it is still not easy to suction microparticles that are smaller than 2.0 µm [15]. Moreover, in the visualization experiment that we conducted, not only was the effect of the extraoral vacuum almost invisible, unless it was placed almost directly above the treatment site, but the flow of fine particles rising up to pass through it was still observed (data not shown). Thus, completely suppressing the generated aerosol particles was impossible.

Bruna et al. [25] argued that the use of rubber dams should be mandatory, and Butera et al. [26] minimized the use of aerosol-producing instruments, such as ultrasound and air, to reduce the production of highly polluted aerosols. It has been reported that it is necessary to promote the use of chlorhexidine, hydrogen peroxide, or povidone iodine; the development of new treatments, such as laser and ozone; and the administration of probiotics prior to each dental treatment [26,27].

Rexhepi et al. [16] also emphasized the importance of ventilation. They performed an experiment with 14 steps with and without natural ventilation in the dental unit, considering PM1, PM2.5, PM10, and breathable (<4 µm) categories. The total dust was measured. The total PM levels were higher during scaling than during other procedures (the data suggest that natural ventilation, with both windows and doors open, is not adopted during dental treatment). Further, it has been demonstrated that the use of standard saliva drainers can significantly reduce the total release of PM10 [20].

Ehtezazi et al. [28] used a phantom head as a model to measure the size and number of particles in a dental clinic using a high-resolution, electrical low-pressure impactor particle sizer at six points around them. Dental aerosol-generating procedures produce aerosols characterized by particles with a diameter of <0.3 μm. Although aerosol removal interventions, such as high-volume intraoral suction alone or in combination with an air cleaning system, may rapidly reduce particle concentrations to within background ranges, in our preliminary experiments, the intraoral vacuum did not show a sufficient effect. Thus, additional experiments are required to elucidate this discrepancy [28].

It is also common in dental clinics for multiple patients to undergo treatment simultaneously, resulting in a very high density of aerosol particles in treatment rooms when high-speed cutting tools are being used. Thus, the longer the dwelling time, the greater the risk of exposure for others. Some aerosol particles are assumed to attach to the floor and to objects in the room, including desks, personal computers, and surfaces of the human body. Thus, there is not only a risk of airborne infection, but attached aerosol particles may also lead to contact infections. SARS-CoV-2 survived up to 72 h on stainless steel and plastic surfaces, up to 4 h on a copper surface, and up to 24 h on a cardboard surface. In addition, the virus can survive for up to 1 day on clothing or up to 7 days on the outer layer of surgical masks. It also exhibits various stability depending on the ambient temperature, withstanding 4 °C for up to 14 days, 22 °C for 7 days, 37 °C for 1 day, and 56 °C for up to 10 min [22,29]. In addition, devising the order and process of appointment reservations is important so that the waiting room is not crowded and to allow enough time for necessary patient interactions to be performed after an examination is completed. Telemedicine and triage are also powerful tools for assessing patients’ conditions before they arrive at dental clinics and avoiding the presence of sick patients [25].

Therefore, there seems to be a high risk of COVID-19 infection due to aerosols from dental care, although there are few reports of dental care-related COVID-19 infection or the occurrence of clusters. Because of the large amounts of water that are used when high-speed rotating cutting tools are employed, the effects of dilution may imply that the aerosols generated contain almost no SARS-CoV-2 virus, or they may contain such small quantities such that no infections occur. Having the patient wear a mask or face shield does reportedly make it possible to curb the risk of infection effectively, but this approach would not be feasible in dental therapy. In simulations, combining face masks and face shields reportedly provides 63% to >99.9% protection against infection in medical practice [30]. Therefore, using personal protective equipment such as masks, face shields, or aprons, is important for preventing infections [11,31,32]. Dental practitioners should implement fundamental infection controls, such as taking standard precautions [11,12,13,17,31,32,33], diligently setting up ventilation [16], and using air purifiers. They should also consider using other devices in treatment rooms and improving the environment, such as cleaning chairs that are used for treatment. While excellent PPE is ideal, not all medical facilities can achieve the required standard. Dental practitioners should take standard precautions, e.g., the installation of equipment, such as comprehensive ventilation systems and air purifiers in clinical settings, and clean patient treatment chairs. It is desirable to take basic infection control measures, such as maintenance. Further, the “three cs” advocated by the Japanese government, which provide an easy-to-understand reminder that can be understood well by the general public on what should be avoided (i.e., 1: closed spaces with poor ventilation; 2: crowded places with many people nearby; 3: close-contact settings, such as close-range conversations) are important.

In this study, we aimed to accurately reproduce the context of dental treatment, but it could not be completely simulated. First, we used a circular bowl with a diameter of 8 cm and a depth of 5 cm that imitated the oral cavity. We did not use a vacuum. The experimental setup was suitable for investigating the formation of particulates under certain conditions. However, in actual human treatment, the size of the oral cavity varies per person. Moreover, various scattering patterns are possible depending on the treatment site and the position of vacuum device use, and it is impossible to reproduce all of them. Second, the dental equipment used is also relevant. A device called an air turbine handpiece is popularly used for dental treatment. It is estimated that rotation speed reaches 300,000 to 400,000 rpm, requires more water and air, and has more fine particles scattered. In the visualization experiment using laser light that we conducted prior to this experiment, the air turbine handpiece generated a large air flow that could be said to be a larger water column. However, it was not possible to use this equipment in a SCL because only portable devices could be brought in. Furthermore, there was a possibility that accurate fine particles could not be measured due to a large amount of air flow generated from the portable devices, especially during and after use. The air turbine handpiece, because it originally has less torque during cutting, requires an air compression mechanism and is inconvenient to carry. Because the performance of the micromotor used in this research has improved significantly recently, an air turbine handpiece would not need to be used in the future. Therefore, it seems that the simulation using only this device was an appropriate choice considering the usage situation of the device in the near future. Third, in this study, the influence of fine particles that could be background particles originally floating in the space can be removed. Therefore, the amount, range, time, etc. of fine particles actually suspended in the air can be measured when using a high-speed motor with extreme accuracy. The measurement environment was in a closed space, whereas it has been reported that the number of fine particles scattered in an actual dental treatment room is larger than that in other places. In addition, clinics are typically air-conditioned, people are expected to be active, and particle movements can lead to complex trends. This is the limitation of the simulation of this study. However, from the opposite perspective, this study provides an important default experimental result that eliminates the existence of background fine particles that are the basis of all conditions. To the best of our knowledge, the present study is the first to investigate this subject. 

Presently, knowledge of the infectiousness of COVID-19 from the mouth or saliva of a patient remains insufficient, and, thus, more research and information are required. In addition, the infectiousness or stability of viruses that are contained in aerosols generated in the mouth should be investigated in future studies.

## 5. Conclusions

A large number of aerosol particles generated from the use of the micro engine rapidly rose and were suspended in the atmosphere of a room within 30 s. Within a radius of 100 cm of the user, the scattering peaked in about 90 s. Furthermore, it was found that the particles diffused into the room as time passed, reaching a maximum after 210 s, and then gradually decreased, and the 4 × 4 m room returned to equilibrium. We conducted experiments in a super clean laboratory in a closed state without using a vacuum device. Although conditions were not a faithful reproduction of the actual condition of dental clinics, the study obtained a groundbreaking result demonstrating the scattering of fine particles for the first time, without the complexity of various possible backgrounds.

## Figures and Tables

**Figure 1 ijerph-18-11279-f001:**
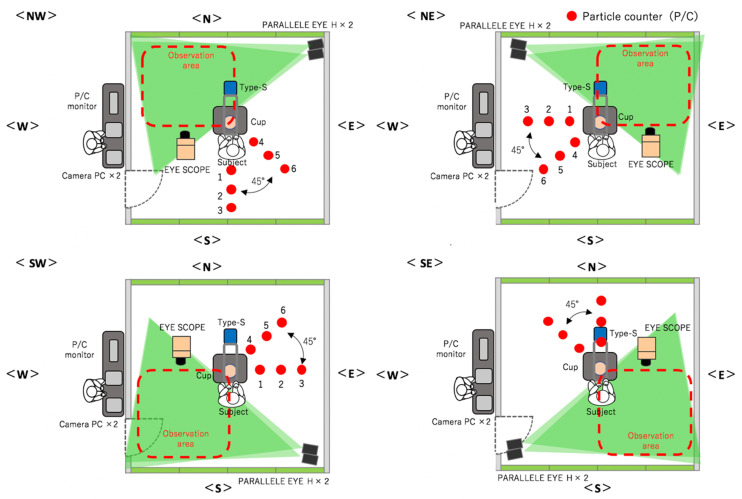
Experimental environment with four conditions (NW, SW, SE, NE) and three rounds each.

**Figure 2 ijerph-18-11279-f002:**
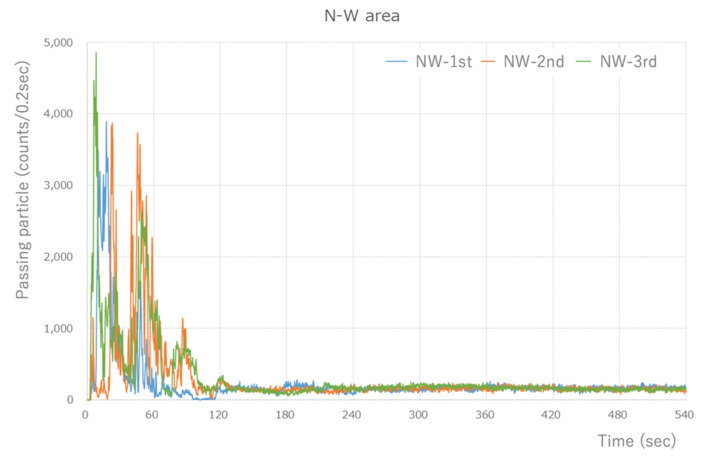
Type-S measurement results (during visualization of the NW area). A large number of particles were detected, at around 4000 counts/0.2 s, immediately after the start. However, the number of particles that were measured after 1 min of work significantly decreased, and a steady state was reached after 60 s. The results are similar for the other areas.

**Figure 3 ijerph-18-11279-f003:**
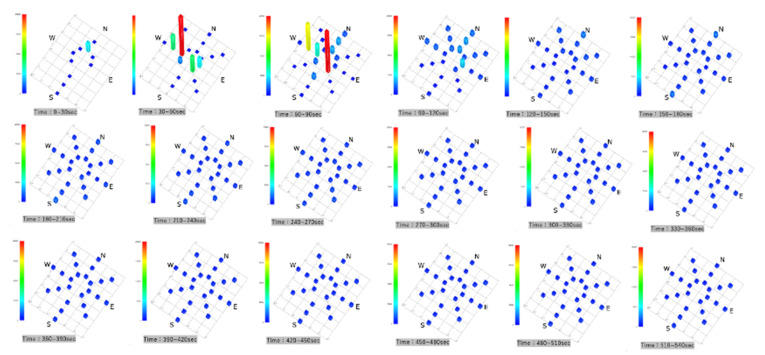
Accumulated images of particles larger than 5 µm visualized with the Parallel Eye H.

**Figure 4 ijerph-18-11279-f004:**
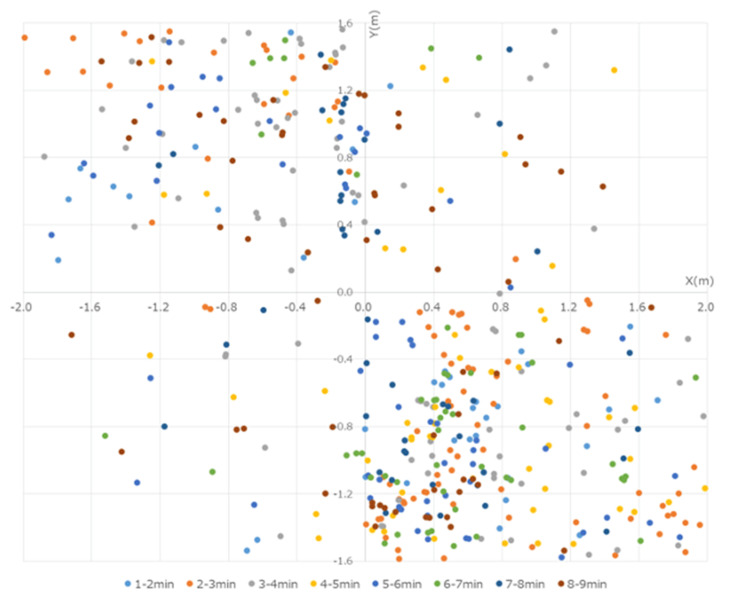
Quantity of particles measured every 30 s with the P/Cs. Immediately after the start of the experiment, many particles were in the 50 cm range around the workers. The particles were distributed westward. By 120 s, the quantity of particles measured at each location reached a similar amount, and the particles spread out in the room over time.

**Table 1 ijerph-18-11279-t001:** Equipment used.

Light sources	Two scanning laser sheet light sources (“Parallel Eye H”)
Cameras	One specialized microparticle-visualizing high-sensitivity camera (“i-Scope”)
Image processing	The basic image processing software package “ParticleEye Viewer”
Particle measurement	One mobile visualizing counter system (“Type-S”) Two types of particle counters (Six units in total) A: Particle Plus model P311 (four units) B: TSI model AeroTrak9110 (two units)

## Data Availability

The data that support the findings of the study are available from the corresponding author (Maruoka, Y.) upon reasonable request.

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
