# Peer review of "Dispersion of Aerosols Generated during Dental Therapy"

_ijerph, 2021, doi:10.3390/ijerph182111279_

Round 1
Reviewer 1 Report
The article entitled “Dispersion of Aerosols Generated During Dental Therapy” is an interesting paper regarding the dynamics of the aerosols produced when high-speed rotating tools are used in dentistry. Is a well structured manuscript and my solely advices were the following:
- In the introduction part the author should discuss other studies ( see doi: 10.3390/ijerph18147472) regarding PM particles distribution during the performarce of dental procedures in daily practice
- The Authors should analyze in the Discussion which procedures were performed the most during COVID-19 pandemic ( see doi: 10.3390/ijerph17165780 and how the current guidelines suggest to minimize the uses of airborne spreading devices in order to reduce the production of PM particles that can act as vectors of infectious agents among patients and dentists
- The conclusion should be more supported by the results.
Author Response
Point 1: In the introduction part the author should discuss other studies ( see doi: 10.3390/ijerph18147472) regarding PM particles distribution during the performarce of dental procedures in daily practice.
Response 1: We added that knowledge. We would please you to refer to the text. Thank you for your advice. Please see the attachment.
Point 2: The Authors should analyze in the Discussion which procedures were performed the most during COVID-19 pandemic ( see doi: 10.3390/ijerph17165780 and how the current guidelines suggest to minimize the uses of airborne spreading devices in order to reduce the production of PM particles that can act as vectors of infectious agents among patients and dentists.
Response 2: We added that knowledge. We would please you to refer to the text. Thank you for your advice.
Point 3: The conclusion should be more supported by the results.
Response 3: We accept with your kind remarks. We removed unfounded contents and have rewritten.

Reviewer 2 Report
I congratulate the Authors for all the effort that they did to conduct this study. The topic is very interesting and appreciable from a scientific and clinical point of view. The study is well designed and conducted, and the manuscript is clear. There are some comments below:
The English is clear and spelled correctly
Abstract
The abstract correctly summarizes the study design and purpose.
Keywords
The keywords are correct and perfectly fitting the study design.
Introduction
The introduction is well organized and clear.
Materials and Methods
The methodology is well described and complete. The sample size and statistical procedures are correct.
Results
The results are clear and supported by an adequate number of figures and tables.
Discussion and Conclusions
The discussion is well conducted and justified by the literature,I would add in the discussions a reference related to aerosol reduction during non-surgical periodontal therapy procedures:
Bio-inspired systems in nonsurgical periodontal therapy to reduce contaminated aerosol during COVID-19: A comprehensive and bibliometric review
this link is disabled,
The conclusions are correct.
Author Response
Point 1:
The discussion is well conducted and justified by the literature,I would add in the discussions a reference related to aerosol reduction during non-surgical periodontal therapy procedures:
Bio-inspired systems in nonsurgical periodontal therapy to reduce contaminated aerosol during COVID-19: A comprehensive and bibliometric review
,
We added that knowledge. We would please you to refer to discussion. Thank you for your useful advice. Please see the attachment

Reviewer 3 Report
Point 01
“The aerosol particles that were generated by running the micro engine included only a few particles that were >5 μm in size. In addition, the range of these particles was within 50 cm of the patient, and they were no longer measured when the equipment was stopped. For the particles that rose up to large heights, the visualized particles descended from 30 s onward. Based on the qualitative assessment, they were evenly distributed and dispersed everywhere. More aerosol particles that were less than 5 μm in size were generated than those that were larger than 5 μm in size, and they spread upward. They were first distributed to the left of the workers but gradually spread out everywhere and permeated the room once 2 min had elapsed.”
The third paragraph of the Discussion (above) is a pure repetition of the Results.
Point 02
Paragraphs 4-7 of the Discussion is a review of the literature without an actual discussion of the findings of the study.
Point 03
In the Discussion there is only a discussion of the methodology applied in the study (paragraph 8 of the Discussion). An actual discussion of the results of the study is still lacking. There is virtually no discussion of the findings.
Point 04
“In the experiment described in this paper, laser light, a high-sensitivity camera, and P/Cs were utilized to examine how droplets and aerosols produced by a micro engine are transported and spread out.”
This is not a conclusion of your study. This seems like the Materials and Methods of an abstract.
Point 05
“This finding underscores the importance of ventilation because treatments are often performed in parallel in dental clinics. Although the infectiousness of COVID-19 could not be assessed, many aerosol particles with the same size were generated, and there have been no reports of infections or clusters at dental clinics.”
This is not a conclusion of your study. This is an opinion (first sentence) followed by a non-verified assumption (second sentence).
Point 06
“This experiment suggests that COVID-19 infection is very unlikely to occur from the aerosols that are generated during dental care.”
This is not a conclusion of your study. This is an assumption that was not verified. Moreover, COVID-19 infection may be influenced not only by how the aerosols propagate through the dental office, but by other factors as well.
Author Response
Point 01: “The aerosol particles that were generated by running the micro engine included only a few particles that were >5 μm in size. In addition, the range of these particles was within 50 cm of the patient, and they were no longer measured when the equipment was stopped. For the particles that rose up to large heights, the visualized particles descended from 30 s onward. Based on the qualitative assessment, they were evenly distributed and dispersed everywhere. More aerosol particles that were less than 5 μm in size were generated than those that were larger than 5 μm in size, and they spread upward. They were first distributed to the left of the workers but gradually spread out everywhere and permeated the room once 2 min had elapsed.”
The third paragraph of the Discussion (above) is a pure repetition of the Results.
Response 1: We deleted the part as you pointed out. We appreciate your advice.
Point 2: Paragraphs 4-7 of the Discussion is a review of the literature without an actual discussion of the findings of the study.
Response 2: We added that knowledge. You would please you to refer to the text. Thank you for your useful advice.
Point 03: In the Discussion there is only a discussion of the methodology applied in the study (paragraph 8 of the Discussion). An actual discussion of the results of the study is still lacking. There is virtually no discussion of the findings.
Response 3: We added that knowledge. You would please you to refer to the text. Thank you for your useful advice.
Point 04: “In the experiment described in this paper, laser light, a high-sensitivity camera, and P/Cs were utilized to examine how droplets and aerosols produced by a micro engine are transported and spread out.”
This is not a conclusion of your study. This seems like the Materials and Methods of an abstract.
Response 4: We added that knowledge. You would please you to refer to the text. Thank you for your useful advice.
Point 05: “This finding underscores the importance of ventilation because treatments are often performed in parallel in dental clinics. Although the infectiousness of COVID-19 could not be assessed, many aerosol particles with the same size were generated, and there have been no reports of infections or clusters at dental clinics.”
This is not a conclusion of your study. This is an opinion (first sentence) followed by a non-verified assumption (second sentence).
Response 5: We added that knowledge. You would please you to refer to the text. Thank you for your useful advice.
Point 06: “This experiment suggests that COVID-19 infection is very unlikely to occur from the aerosols that are generated during dental care.”
This is not a conclusion of your study. This is an assumption that was not verified. Moreover, COVID-19 infection may be influenced not only by how the aerosols propagate through the dental office, but by other factors as well.
Response 6: We added that knowledge. You would please you to refer to the text. Thank you for your useful advice.
Your precise advice has reminded us of the importance of this research.
Please see the attachment

Reviewer 4 Report
This research is under the scope of this journal; the topic is relevant for readers, and this research deals with potentially significant knowledge to the field.
However, there are some concerns in the about the present manuscript:
Abstract
- It must be reformulated, following the indications for the realization of the abstract, without the words: Aim, methods and results, and conclusion.
Introduction
- What is the importance of this study for dental clinical? What is the gap on this field of literature?
- It necessary change some procedures on dental office? Which results are comparable with others study in the oral care? For support infection control measures for implemented in dental offices, we recommend a reference to support (COVID19-NOC | PDF - Rapid Guidelines on COVID 19 for Dentistry with a level of evidence for the existing bibliography and with a degree of recommendation. It’s were based in National Institute for Health and Care Excellence (www.NICE.org) ISBN-13 (15) 978-989-26-2072- 5 doi https://doi.org/10.14195/978-989-26-2072-5; (EN-version) http://monographs.uc.pt/iuc/catalog/view/28/343/627-1).
- What was the null hypothesis for this study?
Methods
- The authors can made a statistical analysis
Results
- Did authors perform power analysis? What was sample size appropriate to evaluate this aim of this study?
Discussion
- Please, clarified what was the limitation of this study?
And also, clarified the future perspectives also add in the discussion.
Author Response
Point 1: It must be reformulated, following the indications for the realization of the abstract, without the words: Aim, methods and results, and conclusion.
Response 1: As you advised, we have rewritten the content of this treatise to be clear.
Point 2: What is the importance of this study for dental clinical? What is the gap on this field of literature?
Response 2: We are conducting a preliminary experiment prior to this study. In the preliminary experiment, it was strange that the fine particles generated from the dental treatment equipment floated irregularly over a wide area for a long time without being proportional to the distance. We wondered how aerosol particles are generated, how they spread, how long they float in space, and how many and large particles are present. Therefore, we tried to feature of the fine particles, especially in a super clean laboratory in an environment excluding all background effects. We believe that is the greatest attraction of this paper.
Point 3: It necessary change some procedures on dental office? Which results are comparable with others study in the oral care? For support infection control measures for implemented in dental offices, we recommend a reference to support (COVID19-NOC | PDF - Rapid Guidelines on COVID 19 for Dentistry with a level of evidence for the existing bibliography and with a degree of recommendation. It’s were based in National Institute for Health and Care Excellence (www.NICE.org) ISBN-13 (15) 978-989-26-2072- 5 doi https://doi.org/10.14195/978-989-26-2072-5; (EN-version) http://monographs.uc.pt/iuc/catalog/view/28/343/627-1).
Response 3: As we said in the previous section, other similar studies could not completely eliminated the effect of background. This research investigating the trends of fine particles generated only from dental treatment equipment is unprecedented as far as we concerned, and this seems to be a point that can differentiate it from other research.
Point 4: What was the null hypothesis for this study?
Response 4: We finally noticed from your point, but we did not set any clear null hypothesis although there were items we wanted to investigate. Thank you for your guidance. We wanted to deny the result of this kind of research, which states that "fine particles generated by dental treatment equipment are suspended only for a short time and are not scattered over a very wide area."
Point 5: The authors can made a statistical analysis.
Response 5: We are very sorry, but we do not perform statistical analysis. We measured the fine particles generated from the micro engine three times, but the tendency was not constant and it was judged to be unsuitable for statistical processing. However, the value of the particle counter adopts the median value.
Point 6: Did authors perform power analysis? What was sample size appropriate to evaluate this aim of this study?
Response 6: Power analysis could not be performed. This experiment required a great deal of money and effort. Also, since it is not possible to measure the entire clean room at once, it is divided into 4 areas for three times. Since it was expected that a total of 15 or more measurement experiments would be completed from early morning to night, including preliminary experiments, it was impossible to set a sample size of 3 or more from the beginning.
Point 7: Please, clarified what was the limitation of this study?
Response 7: Since our experiments were conducted in a super clean laboratory, we were unable to completely reproduce the form of normal dental practice. In other words, faithfully the generation of aerosols in the actual dental clinic as in the paper of Imena et al. (Int. J. Environ. Res. Public Health 2021, 18, 7472. https://doi.org/10.3390/ijerph18147472) It was not possible to reproduce, and it was not possible to search for the possibility that the floating fine particles would be zero.
Point 8: And also, clarified the future perspectives also add in the discussion.
Response 8: Following your valuable advice, we added the following sentence to the discussion:
Please see the attachment

Round 2
Reviewer 3 Report
The manuscript now seems to be suitable for publication.
Reviewer 4 Report
This research is under the scope of this journal; the topic is interesting for readers and this research deals with potentially significant knowledge to the field and an open new way for future studies.
The authors improved the quality of the manuscript after the reviewer's indications.